# Inflammasome Activation in Retinal Pigment Epithelium from Human Donors with Age-Related Macular Degeneration

**DOI:** 10.3390/cells11132075

**Published:** 2022-06-30

**Authors:** Mara C. Ebeling, Cody R. Fisher, Rebecca J. Kapphahn, Madilyn R. Stahl, Shichen Shen, Jun Qu, Sandra R. Montezuma, Deborah A. Ferrington

**Affiliations:** 1Department of Ophthalmology and Visual Neurosciences, University of Minnesota, Minneapolis, MN 55455, USA; ebeli017@umn.edu (M.C.E.); fishe765@umn.edu (C.R.F.); kapph001@umn.edu (R.J.K.); stahl154@umn.edu (M.R.S.); smontezu@umn.edu (S.R.M.); 2Graduate Program in Biochemistry, Molecular Biology, and Biophysics, University of Minnesota, Minneapolis, MN 55455, USA; 3Department of Pharmaceutical Sciences, SUNY Buffalo, Buffalo, NY 14203, USA; shichens@buffalo.edu (S.S.); junqu@buffalo.edu (J.Q.); 4Doheny Eye Institute, Pasadena, CA 91103, USA

**Keywords:** age-related macular degeneration, complement factor H, inflammasome, inflammation, retinal pigment epithelium

## Abstract

Age-related macular degeneration (AMD), the leading cause of blindness in the elderly, is characterized by the death of retinal pigment epithelium (RPE) and photoreceptors. One of the risk factors associated with developing AMD is the single nucleotide polymorphism (SNP) found within the gene encoding complement factor H (CFH). Part of the innate immune system, CFH inhibits alternative complement pathway activation. Multi-protein complexes called inflammasomes also play a role in the innate immune response. Previous studies reported that inflammasome activation may contribute to AMD pathology. In this study, we used primary human adult RPE cell cultures from multiple donors, with and without AMD, that were genotyped for the Y402H CFH risk allele. We found complement and inflammasome-related genes and proteins at basal levels in RPE tissue and cell cultures. Additionally, treatment with rotenone, bafilomycin A, and ATP led to inflammasome activation. Overall, the response to priming and activation was similar, irrespective of disease state or CFH genotype. While these data show that the inflammasome is present and active in RPE, our results suggest that inflammasome activation may not contribute to early AMD pathology.

## 1. Introduction

Age-related macular degeneration (AMD) is characterized by central vision loss, as a result of retinal pigment epithelium (RPE) and photoreceptor cell death in the macular region of the retina [1,2]. The RPE cell layer, located between the retinal photoreceptors and the outer retinal blood supply of the choroid, fulfills many key functions in the eye. These functions include phagocytosis of the shed photoreceptor outer segments, transport of nutrients and oxygen from the choroid to the outer retina, and secretion of factors that are crucial for the health and integrity of the retina and choroid [3].

Many risk factors contribute to AMD pathogenesis, including advanced age, genetic predisposition, and environmental stressors, such as smoking and a high fat diet [4]. There are a number of single nucleotide polymorphisms (SNPs) associated with increased risk of developing AMD [5]. One of these SNPs, found in ~50% of AMD patients, is located at position 1277 in the gene encoding complement factor H (CFH), where there is a nucleotide switch from T to C (rs1061170) [6]. This SNP results in an amino acid substitution of tyrosine to histidine at position 402 (Y402H) in the CFH protein. The single amino acid change is biologically and clinically important, in that individuals harboring the high-risk C allele that produces the Y402H CFH protein have a 3- to 5-fold increased likelihood of developing AMD [7].

CFH is the main inhibitor of the alternative complement pathway, which is part of the innate immune system that kills invading pathogens by lysing pathogenic cells. Extracellular CFH protects host cells from inappropriate attack by the complement system, preventing lysis of host cells and chronic inflammation [8]. CFH protection comes from the inhibition of C3 conversion into its active products C3a and C3b [9]. Overproduction of C3a and C3b leads to the cleavage of C5 into C5a and C5b, and subsequent formation of the membrane attack complex (MAC) on the cell’s membrane. The MAC leads to cell lysis and chronic inflammation. A recently described role for intracellular CFH shows it acts as a cofactor for cathepsin L, which cleaves C3 into C3a and C3b [10,11]. Prior analysis of the CFH Y402H risk variant demonstrated it has reduced ability to inhibit complement activation, thereby leading to increased C3a and C3b, potentially contributing to the strong association between chronic inflammation and AMD initiation and progression [12].

Activation of the innate immune response, and resultant inflammation, is triggered by exogenous (e.g., microbes or toxins) or endogenous (e.g., ATP or DNA) danger signals that are identified by pattern recognition receptors (PRR). Some of these PRR include toll-like receptors (TLRs), nucleotide-binding oligomerization domain-like receptors (NLRs), and absent in melanoma 2-like (Aim2) [13]. PRRs can initiate the formation of multi-protein complexes known as “inflammasomes”. These complexes consist of the PRR (the scaffolding protein), ASC (the adaptor protein), and an effector caspase (pro-caspase-1). Inflammasome activation requires two signals, an initial priming signal and a second activating signal.

Figure 1 shows the steps involved in the assembly and activation of the inflammasome. In this study, primary adult human RPE cell cultures (haRPE) were first primed with IL-1α and LPS, molecules that activate the NF-κB pathway, which upregulates the transcription of Pro-IL-1β and complement pathway genes. Formation of inflammasomes was induced by reagents tied to purported AMD disease mechanisms. Rotenone was used to induce mitochondrial DNA (mtDNA) damage and mitochondrial dysfunction [14,15,16]. Bafilomycin was used to inhibit lysosomes and induce buildup of damaged organelles within haRPE cells [17]. Rapid release of ATP from damaged RPE cells could enter adjacent cells via the P2X7 receptors and activated the inflammasomes [18,19]. Inflammasome assembly initiates Caspase-1 cleavage, and maturation and secretion of the inflammatory cytokine, Interleukin-1β (IL-1β) [20].

As described above, both extracellular and intracellular complement pathways can initiate inflammasome assembly [21,22]. Additionally, the indirect action of the complement cascade via engagement of complement receptors, C3aR and C5aR1, initiates a broad range of signaling pathways, including NF-κB and mammalian target of rapamycin (mTOR), which could then induce inflammasome activation [22].

Previous studies have reported inflammasome activation in RPE, although the contribution of inflammasomes to AMD pathology has not been clearly defined. In macular RPE tissue from donors with geographic atrophy that is associated with late AMD, an increase of NLRP3, ASC, Caspase-1, IL-1β, and IL-18 proteins was observed [23,24,25]. However, another study reported NLRP3 inflammasome and the downstream cytokine IL-18 had a protective role in AMD [26]. This study addresses the controversy surrounding the conflicting evidence for the role of inflammasome activation in AMD pathogenesis, by comparing the response of haRPE cells with and without AMD. We also examined whether the presence of the CFH Y402H risk allele altered inflammasome activation. The results show treatment with rotenone, Bafilomycin A, and ATP led to activation of the inflammasome. Overall, the response to priming and activation was similar, irrespective of disease state or CFH genotype. While these data show that the inflammasome is present and active in RPE, the results suggest that inflammasome activation may not contribute to early AMD pathology.

## 2. Materials and Methods

### 2.1. Human Eye Procurement and Grading for AMD

De-identified donor eyes were obtained from the Lions Gift of Sight (Saint Paul, MN). Eyes were obtained with written consent of the donor or donor’s family for use in medical research, in accordance with the Declaration of Helsinki. The Lions Gift of Sight is licensed by the Eye Bank Association of America (accreditation #0015204) and accredited by the FDA (FDA Established Identifier 3000718538). De-identified donor tissue is exempt from the process of Institutional Review Board Approval.

Tissue handling, storage, and donor exclusion criteria were as outlined previously [27]. Evaluation of the presence or absence of AMD was determined by a Board Certified Ophthalmologist (Sandra R. Montezuma) from stereoscopic fundus photographs of the RPE, using the criteria (RPE pigment changes and presence, drusen size and location) established by the Minnesota Grading System [28]. Records from the Lions Gift of Sight provided demographics (age, gender, cause of death, time to tissue processing) for the donor tissue used in proteomic analysis (Appendix A) and to generate primary cultures of RPE cells (Appendix A).

### 2.2. Complement Factor H Y402H Genotyping

Genomic DNA was extracted from graded donor retinal tissue using a QIAmp DNA micro kit (Qiagen, Valencia, CA, USA). The extracted DNA was quantified using a Quant-iT PicoGreen dsDNA Assay kit (Life Technologies, Carlsbad, CA, USA). Subsequent PCR amplification and sequencing was performed as previously described [29].

### 2.3. Sample Preparation and Nano Liquid Chromatography (LC)-Mass Spectrometry (MS/MS) Analysis

Demographics of the donors used in the RPE protein analysis performed using Mass Spectrometry are found in Appendix A. All donors used had the CFH CT genotype. To isolate an organelle-enriched fraction, RPE cell pellets were suspended in isolation buffer (70 mM sucrose, 200 mM mannitol, 1 mM EGTA, 10 mM HEPES pH 7.4) and subjected to two freeze/thaw cycles prior to homogenization. Samples were centrifuged at 800× *g* for 8 min. The supernatant containing the organelles was centrifuged at 12,000× *g* for 10 min. The pellet containing the organelle-enriched fraction was resuspended in ice-cold Tris buffered saline and frozen at −80 °C.

All samples were extracted with an ice-cold surfactant cocktail buffer (50 mM Tris-formic acid, 150 mM NaCl, 0.5% sodium deoxycholate, 2% SDS, 2% IGEPAL CA630, pH 8.0), as described [30,31]. The same amount of protein (60 μg) was processed using an acetone precipitation/on-pellet-digestion approach. The peptide mixture of 4 μg was separated using a nano-LC system (Eksigent, Dublin, CA, USA) and analyzed using an Orbitrap Fusion mass spectrometer (Thermo Fisher Scientific, Waltham, MA, USA). A nano-LC column (75 μm ID × 65 cm, packed with 3-μm particles) with a 160-min gradient was used for separation. The MS data were acquired using the data-dependent product ion mode with a cycle time of 3 s. MS1 survey scans (*m*/*z* range 400–1500) were acquired using Orbitrap at a resolution of 240,000, with an automatic gain control (AGC) target of 175%. Precursor ions were isolated by the quadrupole with a 1.6-Th window and fragmented by high energy collision dissociation (HCD). MS2 scans were acquired by Orbitrap at a resolution of 15,000, with a AGC target of 100% and a maximum injection time of 22 ms. Dynamic exclusion was enabled with the following settings: repeat count 1; mass tolerance ±10 ppm; exclusion duration 60 s.

For database searching, LC-MS raw files were searched against the Swiss-Prot *Homo sapiens* protein sequence database (20,304 protein entries, released in October 2020) using MS-GF+ (v2021.01.08), and using the following searching parameters: (1) Precursor ion mass tolerances: 20 ppm; (2) Instrument type: Q-Exactive; (3) Matches per spectrum: 1; (4) Maximal missed cleavages: 2 per peptide; (5) Fixed modification: carbamidomethylation of cysteines; (6) Dynamic modifications: oxidation of methionines and acetylation of peptide N-terminals. IDPicker (v3.1.22706) was used for peptide validation, protein inference/grouping and global false discovery rate (FDR) control, with the following parameters: (1) Protein/peptide FDR: 1%; (2) Maximal protein per protein group: 50; (3) Minimal unique peptide # per protein: 2. UHR-IonStar was used to acquire protein MS intensities based on peptide ion currents [32,33].

### 2.4. Cell Culture

Primary human adult RPE (haRPE) cells were isolated from donor eyes and cultured as previously described [14]. RPE cells (250,000 cells/well, passage 2 or 3) were seeded into Primaria 6-well plates (Corning, Lowell, MA, USA) and allowed to grow for at least 1 month. For treatments, media was changed to 1% FBS. Cells were primed with 5 ng/mL Interleukin-1α (IL-1α) (R & D Systems, Inc., Minneapolis, MN, USA) and 500 ng/mL lipopolysaccharide (LPS) (Sigma, St. Louis, MO, USA) for 16 h. Cells were then treated with 5 µM rotenone (Sigma), 0.2 nM Bafilomycin A (Sigma), or 5 mM ATP (Sigma) to activate inflammasomes for 24 h.

### 2.5. Immunofluorescence

haRPE cells were grown in an 8-well chamber slide (Nunc; Roskilde, Denmark) for one month. Paraformaldehyde-fixed cells were blocked for one hour in 10% normal donkey serum and then incubated in primary antibody (see Appendix A) overnight. After washing with PBS, cells were incubated with secondary antibody for 2 h. Cells were then cover slipped with mounting medium containing 4′,6-diamino-2-phenylindole (DAPI) (Vector Laboratories; Burlingame, CA, USA) and imaged with an inverted confocal microscope, Olympus FluoView FV1000 (Olympus America Inc.; Center Valley, PA, USA).

### 2.6. Western Blotting

Cells were washed with PBS and whole cell lysates were collected after 24-h inflammasome activation. Protein concentration was determined using a BCA assay kit (Thermo Fisher, Waltham, MA, USA). Fifteen micrograms of protein was resolved on SDS-Page gels and transferred to PVDF membrane (Millipore, Burlington, MA, USA). Blots were blocked with 5% non-fat dry milk for 1 h and incubated with primary antibodies (see Appendix A) overnight. After secondary antibody incubation, blots were developed using SuperSignal Western Blot substrate (Thermo Fisher Scientific) and imaged using a ChemiDoc XRS (BioRad, Hercules, CA, USA). Densitometry of band size was quantified using Quantity One software (BioRad). β-Actin was used as a loading control.

### 2.7. ELISA

For cell characterization, haRPE cells (4 × 10^4^ cells/well) were seeded onto 6.5-mm transwell inserts (Corning; Tewksberg, MA, USA). After culturing for 5 weeks, samples were collected from apical and basal chambers 24 h after media change. For inflammasome experiments, media from treated and untreated cells was collected from 6-well plates 24 h after activation. ELISAs for Pigment Epithelium-Derived Factor (PEDF) (R & D Systems; Minneapolis, MN, USA), Vascular Endothelial Growth Factor-A (VEGF-A) (eBioscience; San Diego, CA, USA), Interleukin-1β (IL-1β) (eBioScience), IL-6 (BD Bioscience; San Jose, CA, USA), and C3a (BD Bioscience) were conducted according to the manufacturer’s protocols. Concentrations of protein were derived from a standard curve and normalized to chamber volume.

### 2.8. RNA Extraction, cDNA Synthesis, PCR

Donor eyes were enucleated 4 h or less from death. One eye was placed directly into RNAprotect cell reagent (Qiagen; Hilden, Germany) and stored at −20 °C until RNA isolation. RNA quality was evaluated post-isolation using an Agilent Bioanalyzer 2100 (Biomedical Genomics Center, U of MN). Fundus imaging was performed on the alternative eye within 6 h post-mortem, to evaluate the presence and severity of AMD.

Total RNA of RPE tissue or RPE cultured cells was prepared using the RNeasy Mini kit (Qiagen). RNA was quantified using Nanodrop (Fisher Scientific; Hampton, NH, USA) and cDNA was synthesized with SuperScript Reverse Transcriptase (Thermo Fisher). Quantitative PCR analysis was performed using 2 ng cDNA, iQ SYBR Green supermix (BioRad) and a BioRad iQ5 Multicolor real-time PCR detection system. The forward and reverse primer pairs used in this study are listed in Appendix A. The geometric mean of housekeeping genes, Cyclophilin G (PPIG), Acidic ribosomal phosphoprotein P0 (ARBP), glyceraldehyde-3-phosphate dehydrogenase (GAPDH), was used to calculate the ΔCt for each gene of interest. To determine fold change relative to No AMD (or CFH low risk), ΔΔCt of each AMD haRPE (or CFH high-risk) line was calculated by subtracting the mean ΔCt of No AMD (or CFH low-risk) haRPE. For treatment responses, fold change values were calculated relative to no treatment. A modified Livak method was used to calculate relative expression using the efficiency of each primer.

PCR products were verified by the appearance of a single band migrating at the correct size on a 2% agarose gel, using a 50-bp ladder as reference. PCR products were visualized under UV light and imaged using a ChemiDoc (BioRad).

### 2.9. Live/Dead Assay

Confluent cells (30,000/well) were plated in 96-well clear bottom black-sided plates (Costar; Corning, NY, USA) for priming and activation. Cells were incubated with NucGreen Dead 488 ReadyProbes Reagent (Invitrogen; Waltham, MA, USA) for 20 min. Then, 20× images were collected using a Cytation 1 (BioTek; Winiiski, VT, USA). Cell viability was calculated as the average number of green positive cells in each condition, relative to a RIPA lysis control well (0% viability).

### 2.10. Detecting Cytosolic mtDNA

Following a protocol found on bio-protocol.org, cells were lysed in 1% NP−40, lysates were collected in microcentrifuge tubes and spun at 13,000 rpm for 15 min at 4 °C [34]. The supernatant was transferred to a new tube and ethanol was added before extracting DNA using a QIAamp DNA Mini kit (Qiagen). Quantitative PCR was run using primers specific for the mitochondrial 16S rRNA region and internal control from the 18S rDNA nuclear genome (Appendix A).

### 2.11. Statistical Analysis

Single outliers were identified and removed from each dataset, after performing Grubb’s test. For real-time PCR analysis, unpaired *t*-tests were run on ddCt values, before being transformed into fold change values for graphing. For ELISAs, unpaired *t*-tests were run to compare No AMD to AMD, CFH low- to high-risk, and no treatment to treatment. For Western blot analysis comparing No AMD and AMD or Low and High Risk, all data were calculated relative to the average No AMD or Low Risk value (Figure 3J,L). Values were log transformed and Student’s *t*-test was used to determine the significance between No AMD and AMD groups. For the Western blot analysis comparing treatment effect, fold change values were log transformed, and one sample *t*-tests used to compare no treatment to treatment. For one-sample *t*-tests, data was tested against a hypothetical mean of 0. Analyses were performed using the statistical software in GraphPad Prism 9 (GraphPad Software, La Jolla, CA, USA). Data are reported as mean ± SEM. * *p* < 0.05 and ** *p* < 0.01 were considered statistically significant and a *p*-value (#) between 0.05 and 0.10 was considered a trend.

## 3. Results

### 3.1. Identification of Complement and Inflammasome Proteins in RPE Tissue

To establish whether complement and inflammasome pathway proteins are found in RPE in vivo, we examined RPE tissue harvested from human donor eyes (See Appendix A for donor demographics). A proteomic screen confirmed, with high confidence, the presence of 23 complement components in organelle-enriched fractions in human RPE tissue from No AMD (*n* = 32) and AMD (*n* = 45) donors (Table 1). Statistical comparison of the mass intensity for each protein found no significant difference in content (data not shown). Although we were not able to detect inflammasome proteins in our proteomic screen, the gene expression of inflammasome components (NLRP1, NLRP2, NLRP3, AIM2, cGAS, caspase-1) was detected in RPE tissue using high-quality RNA (RNA integrity number = 8 ± 0.3, mean ± SEM) (Appendix A).

### 3.2. haRPE Cultures to Investigate Inflammasome Activation

Primary RPE cell cultures were generated from human adult donors. A summary of the clinical information and demographics for the donors used in this study is provided in Appendix A. This study included donors with No AMD (MGS1, *n* = 23) or with early (MGS2, *n* = 22) and intermediate (MGS3, *n* = 17) disease stage, which were combined to form the AMD group. The average age of donors in the No AMD group was 63 ± 12.7 years and in the AMD group it was 74 ± 9.5 years. The No AMD group was comprised of 11 female and 12 male donors, and the AMD group was comprised of 13 female and 26 male donors. Classification of donors based on their CFH genotype provided a comparison of donors harboring the low-risk (TT) versus high-risk (CT and CC) allele. This experimental design allowed us to determine whether disease state or genotype influenced inflammasome activation.

As we have shown previously, haRPE exhibit characteristics of native RPE [14,35]. Confluent primary RPE cell cultures from donors used in this study were pigmented and had a cobblestone appearance (Figure 2A). Confluent haRPE cells also attained proper polarization, as evidenced by the basal and apical localization of Bestrophin and Ezrin, respectively (Figure 2B). Results from the Western blot show haRPE cells expressed RPE specific markers (Figure 2C). Consistent with our previous studies, RPE from donors with or without AMD had a similar shape and pigmentation [14,35]. haRPE grown on transwells secreted pigment epithelium-derived factor (PEDF) preferentially to the apical side of the monolayer, and vascular endothelial growth factor-A (VEGF-A) preferentially to the basolateral side. The content of these growth factors revealed no disease-dependent differences (Figure 2D).

### 3.3. Investigation of Complement Pathway and Inflammasome Markers in haRPE Cells

To corroborate findings of the proteomic study and gene expression screen, we assessed haRPE cells under basal conditions for complement and inflammasome related markers. Transcriptional expression of the alternative complement pathway components (C3; Complement Factor B, CFB; CFH, CFI, CD46, CD55, and the MAC complex inhibitor CD59) and markers of inflammation (Monocyte Chemoattractant Protein (MCP-1), Interleukin-6 (IL-6)) were measured to determine the effect of disease state and CFH genotype (Figure 3A,B).

Comparing expression between disease states revealed no significant differences in the expression of complement-related genes (Figure 3A). Evaluating gene expression data based on genotype showed high-risk haRPE had a significantly lower expression of both transcript variants of *CFH* (0.6 fold, *p* = 0.02) and a non-significant decrease of *CD59* (0.8 fold, *p* = 0.08) compared to low-risk haRPE (Figure 3B). Of note, the mean expression of a few genes had a greater than two-fold change, but the high variability within the AMD and high-risk groups prevented this reaching statistical significance.

To further compare complement and inflammation at basal levels, content of C3a and IL6 secretion was measured by ELISA. We that the found secreted levels of these proteins were similar between No AMD and AMD cells (Figure 3C,D) and between low and high risk cells (Figure 3E,F).

A preliminary screen of untreated RPE cell cultures revealed inflammasome gene expression (Appendix A). PCR products of *Aim2* and *NLRP3* were verified by Sanger sequencing (data not shown). Quantitative analysis of inflammasome genes showed that the expression of the pattern-recognition receptors (Aim2, cGAS, and NLRP3), adaptor protein ASC (encoded by the PYCARD gene), and Caspase-1 (CASP1) were similar in AMD and No AMD cells (Figure 3G). While not significant, cytokine *IL-1β* expression was three fold higher in AMD cells compared to No AMD (*p* = 0.08). Inflammasome gene expression levels were also compared in high-risk cells to low-risk cells (Figure 3H). Although not significant, *cGAS* levels were increased (4-fold) in high-risk cells compared to low-risk cells.

To determine if changes in gene expression translated to changes in inflammasome-related protein content, ELISA and Western blotting was performed. The increase in IL-1β gene expression did not translate to increased IL-1β secretion in AMD cells compared to No AMD cells (Figure 3I). Although Aim2 and Pro-Caspase-1 protein content was increased with AMD, this result was not significant, due to the variability between donors (Figure 3J). When sorting data by genotype, we found no differences in IL-1β secretion (Figure 3K) and Aim2 and Pro-Caspase-1 content (Figure 3L) between low and high risk cells. Of note, even though we were able to detect NLRP3 gene expression in our cells, we were not able to detect NLRP3 protein in our cell cultures using the antibody (Cell Signaling #15101) validated to detect this protein using tissue from NLRP3−/− mice (Appendix A) [36].

### 3.4. Inflammasome Assembly in haRPE Cells after Priming and Activation

To stimulate inflammasome activation, haRPE cells were first primed with IL-1α and LPS and then treated with rotenone, Bafilomycin A, or ATP, to activate inflammasomes. We found that the priming step caused significant increases in gene expression for complement components (C3, C3AR, and CFHv1) and inflammation (MCP1 and IL6) (Figure 4A) and also caused a significant increase in secretion of C3a (Figure 4B) and intracellular C3a (Figure 4C), indicative of complement activation. Of note, treatment with rotenone, Bafilomycin A, or ATP did not affect haRPE cell viability (Appendix A).

### 3.5. Treatment Activates Caspase-1 and Triggers IL-1β Secretion in haRPE Cells

We quantified the content of two inflammasome components, Aim2 and Pro-Caspase-1 via Western blot. After rotenone treatment, a three-fold increase in the content of Aim2 was observed (Figure 5A). Although not statistically significant, rotenone treatment caused a 30% increase in cytosolic content of mtDNA, which correlated with activation of Aim2 (Appendix A). Results are consistent with the mechanism of action for Aim2 recognizing mtDNA. Aim2 content did not change after Bafilomycin or ATP treatment. However, activation with Bafilomycin or ATP significantly increased Pro-Caspase-1 in RPE cells compared to their no treatment controls (Figure 5B).

Since we observed increased expression of inflammasome components, we also checked for the activation of Caspase-1 and its downstream effects. Compared to their untreated controls, activation with rotenone or ATP caused significant increases in cleaved Caspase-1 (Figure 5C). All treatments also potentiated the priming effect, leading to increased pro-IL-1β. This effect was most pronounced in cells treated with ATP, with a 10-fold increase in pro-IL-1β (Figure 5D). In addition, rotenone and Bafilomycin A treatments caused a ~5-fold and 4-fold increase in pro-IL-1β content in RPE cells, respectively (Figure 5D). Downstream of Pro-IL-1β, we measured levels of secreted IL-1β after treatment as a measure of inflammasome activation (Figure 5E). Significant increases in concentration of the secreted IL-1β were detected with each treatment compared to their untreated controls. Overall, the three treatments caused an increase in inflammasome activation.

### 3.6. Comparing Inflammasome Activation in haRPE Cells by Disease State and Genotype

The previous data examined the overall response of haRPE to priming and activating agents, to measure inflammasome content and indicators of activation compared to their no treatment controls. We next asked whether the response differed based on disease state (Figure 6A–E, left) or genotype (Figure 6F–J, right).

In general, the overall response did not differ by disease or genotype. In the No AMD vs AMD groups, exceptions included a larger increase in Aim2 content in No AMD cells compared to AMD cells with rotenone treatment (*p* = 0.10; Figure 6A), a larger increase in Pro-Caspase-1 content in AMD cells compared to No AMD cells with ATP treatment (*p* = 0.02; Figure 6B), and a 4-fold increase in Cleaved Caspase-1 in AMD cells compared to no response in No AMD cells with Bafilomycin treatment (*p* = 0.08; Figure 6C). In the CFH low-risk to high-risk groups, the only significant difference was observed after rotenone treatment in the amount of secreted IL-1β (*p* = 0.04; Figure 6J).

Taken together, these results show that inflammasomes could be activated in cultures of haRPE, but disease state or genotype did not have a significant influence on the degree of response.

## 4. Discussion

In this study, we used RPE tissue and primary RPE cell cultures from donors with and without AMD to investigate complement and inflammasome pathways. After determining basal levels of complement and inflammasome in haRPE cells, we promoted inflammasome activation, by treating IL-1α/LPS primed cells with rotenone, Bafilomycin A, or ATP. Prior works showed these treatments induced mitochondrial dysfunction, inhibited autophagy, and stimulated P2X7 receptors, respectively [16,17,25,26,27,28]. Since donors were graded for the presence or absence of AMD and genotyped for CFH Y402H SNP associated with high risk of developing AMD, we were able to perform a comparison of complement and inflammasome-related markers before and after treatment, to determine disease- and genotype-dependent changes.

In RPE tissue, multiple complement proteins were detected (Table 1). Components of the complement pathway were also present under basal conditions in cultured haRPE (Figure 3). Our findings agree with previous studies that also reported complement proteins in lysates and media of primary RPE and iPSC-RPE cells [29,37,38,39,40,41]. On the other hand, despite our use of multiple RPE cell lines, we found that only IL-1β increased with AMD, a result that conflicts with previous studies that found increased C3, CFB, and CFH in AMD iPSC-RPE cells compared to No AMD iPSC-RPE [38]. Among the complement proteins, only CFH and CD59 decreased in the high risk group, a result that conflicts with a previous study using iPSC-RPE that found increased CFH and decreased C3 in CFH high-risk compared to low-risk cells [41]. Discrepancies in findings between studies may be due to the use of different cells (primary vs iPSC-RPE), different culturing conditions, and cells from the different stages of AMD.

In RPE tissue, we did not detect inflammasome proteins via mass spectrometry (Table 1) but were able to find expression of inflammasome genes via PCR (Appendix A). The inability to detect these proteins via mass spectrometry may be due to their low abundance or loss during the preparation of the organelle-enriched samples. In haRPE cell cultures, we observed basal levels of inflammasome proteins and genes (Figure 3 and Appendix A). Expression of inflammasome components has also been reported in other types of cultured RPE, such as ARPE-19 and primary fetal RPE cells [24,42,43,44,45,46]. However, in line with previous reports, NLRP3 protein content was undetectable in RPE cells [36]. Due to the controversy regarding the presence of NLRP3 in RPE, we utilized an NLRP3 antibody validated using NLRP3 knockout mouse tissue (Appendix A) [36]. We were unable to detect NLRP3 using this antibody, though NLRP3 gene expression was confirmed using Sanger sequencing. These results suggest NLRP3 is expressed in RPE, but protein content may be too low for detection via Western blotting.

As expected, an increase in inflammasome-related proteins in haRPE cells was observed upon priming and activation (Figure 4 and Figure 5). Furthermore, all treatments were effective at stimulating inflammasome formation in haRPE cells. In addition to the pathways targeted in this study, previous reports have shown that inflammasomes in RPE can be activated by inhibition of proteasomal degradation, induction of photooxidative damage, and accumulation of toxic Alu RNA transcripts [42,43,47]. Taken together, these results are consistent with the idea that a number of mechanisms could be responsible for activating inflammasomes.

In addition to NLRP3, AIM2 can serve as the scaffold for inflammasome assembly. While we did not detect NLRP3, AIM2 was readily detected by Western blotting. This finding that may be particularly relevant to AMD since activation of the Aim2 inflammasome can occur upon binding of mtDNA that is released into the cytosol from damaged mitochondria. The observed increase in mitochondrial damage and dysfunction with AMD could allow mtDNA release into the cytosol, thereby stimulating AIM2-based inflammasome assembly. However, additional investigations are needed to confirm if this mechanism occurs in RPE with AMD.

When comparing haRPE based on disease state, we found no consistent differences in the response to agents that stimulate inflammasome activation (Figure 6). These results contrast with previous studies that reported increased IL-1β gene expression and Aim2 protein content in AMD RPE compared to No AMD cells [29,48]. Other reports investigating the role of inflammasome in the pathogenesis of AMD observed an increased content of NLRP3, ASC, Caspase-1, IL-1β, and IL-18 protein in RPE tissue from the macula region of donors with geographic atrophy (GA) compared to non-diseased controls [23,24,25]. Another study found increased gene expression levels of NLRP3 and Pro-IL-1β in the macula tissue of GA donors [49]. While these results suggest that inflammasome activation may play a role in AMD, they are inconsistent with our finding of no disease-dependent effect on inflammasome activation. An important distinction is that our study investigated early (MGS2) and intermediate (MGS3) disease, whereas the other studies used human samples with GA, which is late AMD (MGS4). Taken together, these data suggest that inflammasome activation may not play a role in early disease, but rather may contribute to AMD pathology in late stage disease, when geographic atrophy has begun.

While the relationship between CFH Y402H polymorphism and inflammasome activation in RPE has not been adequately studied, recent findings suggest the high-risk CFH variant influences the complement pathway. In the tissues surrounding the RPE, increased C5a and IL-18 immunoreactivity was observed in Bruch’s membrane and choroid in donors with high-risk CC variant compared to the protective variant [25]. When investigating the relationship between systemic cytokines and complement factor H in patients with AMD, systemic levels of IL-6, IL-18, and TNFα were also increased in patients with the high-risk CFH variant compared with patients with the heterozygous CT or low risk alleles [50]. Our present work is the first study to compare the content of inflammasome-associated proteins in primary haRPE cells, as it relates to CFH genotype of the donor. We found no association between basal levels of inflammasome proteins and CFH genotype. We also found no correlation regarding the extent of inflammasome activation and genotype of the donor. However, additional experiments are needed to understand the role of CFH in activation of inflammasomes in RPE cells, including investigating the kinetics of these pathways, as opposed to the single time point utilized in this study.

Notably, in our study, we had no control over donor tissue availability and demographics, including disease severity, gender, and age. While we strove for a balance between these parameters, the distribution may not always have been equal, potentially influencing our results. To minimize technical variability, identical procedures in handling and processing were used. However, haRPE cultures exhibited heterogeneity in both expression of genes and proteins in the complement and inflammasome pathways. Our results are consistent with a comparison of iPSC-RPE lines generated from 15 donors that found donor-to-donor variability exceeded the variability across different lines from one donor, suggesting that donor-specific genetic or epigenetics are responsible for an individual RPE culture’s observed behavior [51]. Another caveat of our model system is that the haRPE cells used in this study came from donors with early and intermediate stage AMD, and our results may not match results from previous studies due to their use of cells from late stage disease.

## 5. Conclusions

This study utilized haRPE cell cultures from multiple donors, with and without AMD, that were genotyped for the CFH Y402H risk allele. We identified complement proteins in haRPE tissue samples, and confirmed the presence of both complement proteins and inflammasome proteins in haRPE. Priming with LPS and IL1α led to the expected upregulation of inflammasome components. Subsequent treatment with rotenone, bafilomycin A, and ATP led to activation of the inflammasome. Overall, the response to priming and activation was similar, irrespective of disease state or CFH genotype. In conclusion, these data show that inflammasomes are present and active in haRPE, although inflammasome activation may not contribute to early AMD pathology.

## Figures and Tables

**Figure 1 cells-11-02075-f001:**
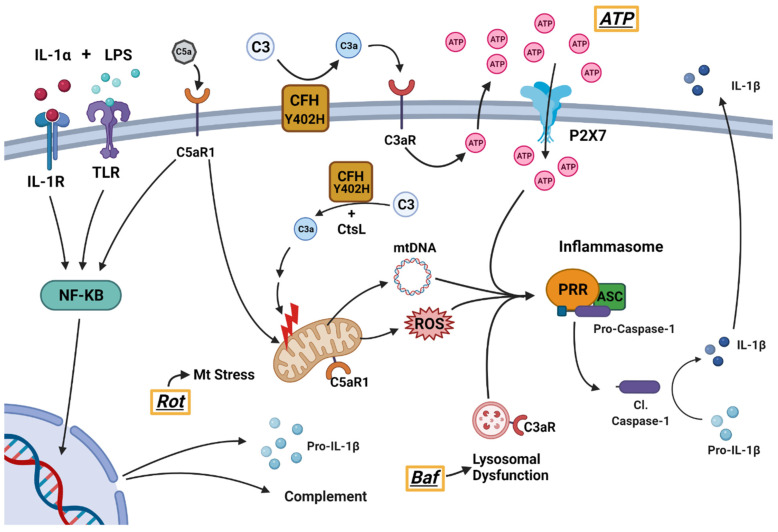
Model of inflammasome activation and complement pathway contributions. Priming via IL-1R and TLR, activated by IL-1α and LPS, leads to NFκB-mediated expression of pro-IL-1β and complement genes. The inflammasome is activated via a number of signals. In this study, we utilized rotenone (Rot) to induce mitochondrial damage, which leads to mtDNA release and generation of ROS. Bafilomycin A (Baf) was used to induce lysosomal dysfunction. ATP was used to stimulate P2X7 receptors. Dysregulation of the complement pathway by CFH Y402H leads to an accelerated turnover of C3 into C3a, which can contribute to mitochondrial damage and activation of C3aR. C3aR activation on the cell’s membrane leads to ATP efflux and subsequent activation of P2X7 receptor. C5aR1 may also contribute to inflammasome pathway activation via NF-κB, as well as mitochondrial damage and intracellular ROS accumulation. These signals ultimately lead to formation and activation of the inflammasome, which is a multi-subunit complex containing a PRR, ASC, and pro-Caspase-1. The assembly of the inflammasome triggers the cleavage of Pro-caspase-1 to cleaved-Caspase-1 (Cl. Caspase-1), which proteolytically activates pro-inflammatory cytokine IL-1β. IL-1β is secreted, leading to inflammation. LPS = lipopolysaccharide, ROS = reactive oxygen species, PRR = pattern recognition receptor, ASC = apoptosis-associated speck-like protein containing a caspase activation and recruitment domain, IL-1β = Interleukin-1β, CSTL = cathepsin L.

**Figure 2 cells-11-02075-f002:**
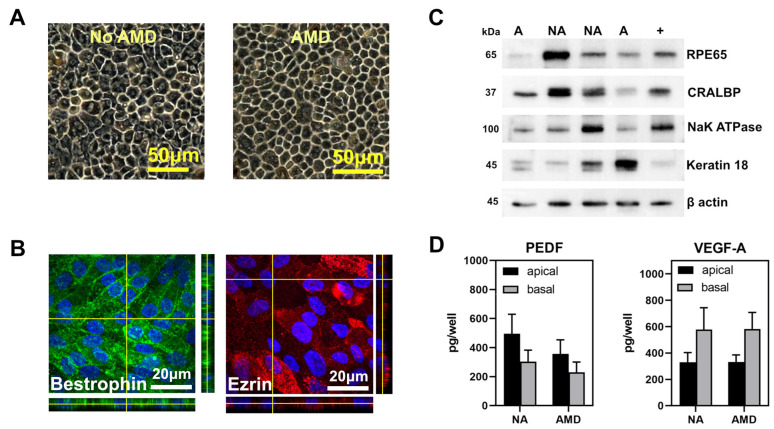
Characterization of haRPE cells. (**A**) Phase microscopy image showing that confluent haRPE form a monolayer with a cobblestone appearance and have pigmentation. Scale bar = 50 um. (**B**) Confocal microscopy images of haRPE. Maximal projection images and orthogonal x-z views showing Bestrophin (green) localized basolaterally and Ezrin (red) localized apically. Nuclei were stained with DAPI (blue). Scale bar = 20um. (**C**) haRPE cultures express prototypic RPE proteins, as demonstrated by Western blots. Molecular mass for each protein is shown on the left. NA = No AMD, A = AMD. (+) is a homogenate of RPE tissue from a human donor. β-Actin is loading control. (**D**) Results from ELISA analysis of pigment epithelium-derived factor (PEDF) and vascular endothelial growth factor A (VEGF-A) content measured in apical and basal media from haRPE grown on transwells.

**Figure 3 cells-11-02075-f003:**
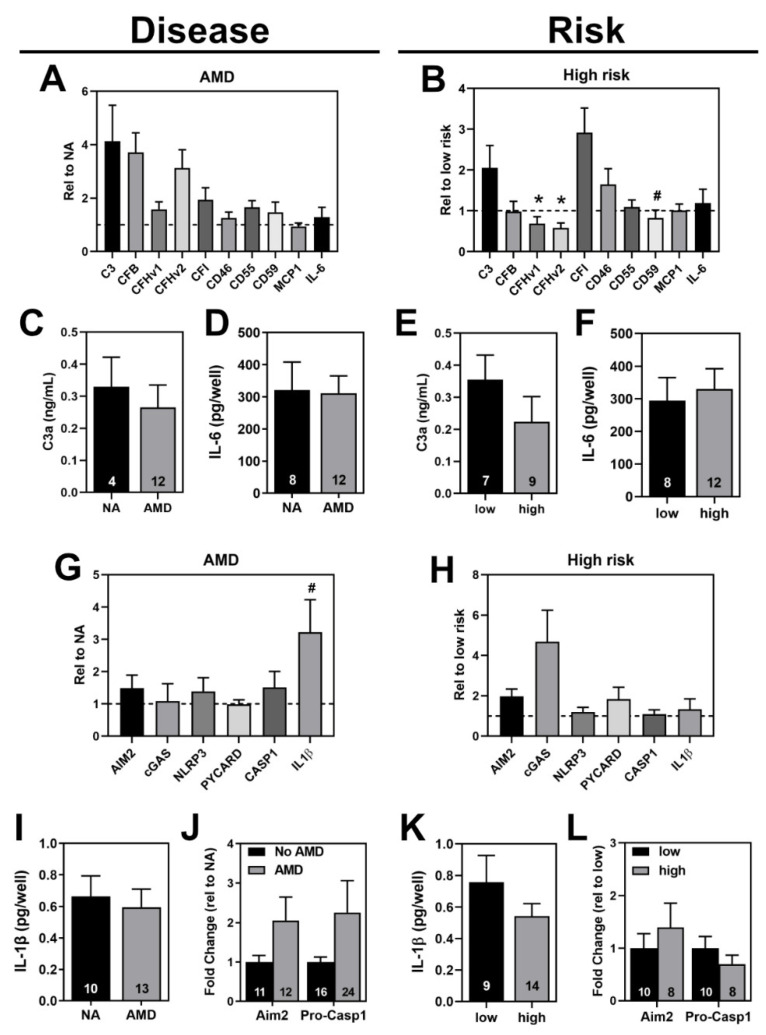
Basal levels of complement and inflammasome in haRPE cells. (**A**,**B**) Gene expression analysis of complement-related genes in haRPE from (**A**) No AMD (*n* = 17) and AMD (*n* = 22) donors or (**B**) CFH low-risk (*n* = 15) and high-risk (*n* = 21) donors. Results shown are fold-change in expression of (**A**) AMD relative to the average of No AMD samples (dashed line) or (**B**) high-risk relative to the average of low-risk donors (dashed line). ELISA for C3a (**C**,**E**) and IL-6 (**D**,**F**) in media from haRPE comparing No AMD and AMD or low-risk and high-risk. (**G**,**H**) Gene expression analysis of inflammasome-related genes in haRPE from (**G**) No AMD (*n* = 6) and AMD (*n* = 6) donors or (**H**) CFH low-risk (*n* = 5) and high-risk (*n* = 7) donors. Results shown are fold-change in expression of (**G**) AMD relative to the average of No AMD samples (dashed line) or (**H**) high-risk relative to the average of low-risk donors (dashed line). ELISA for IL-1β (**I**,**K**) in media from haRPE comparing No AMD and AMD or low-risk and high-risk. (**J**,**L**) Content analysis of inflammasome proteins in (**J**) No AMD and AMD cells or (**L**) low-risk and high-risk cells. Data shown are mean ± SEM. # *p* < 0.1, * *p* < 0.05 determined by unpaired *t* test. Numbers within the bars indicate number of individual donors used in the assay.

**Figure 4 cells-11-02075-f004:**
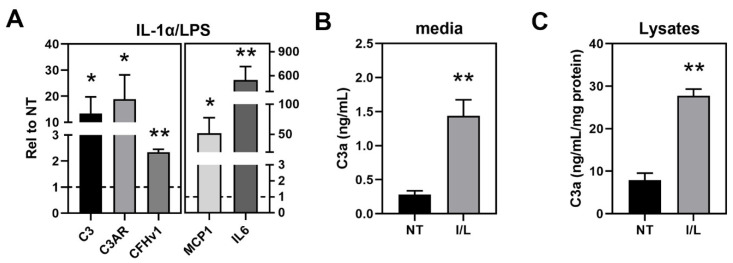
Priming (IL-1α and LPS) induces complement and inflammation gene expression. (**A**) Gene expression analysis of complement-related genes in haRPE cells (*n* = 3) treated with IL-1α + LPS. Results shown are fold-change in expression relative to no treatment. NT = 1 (dashed line). (**B**) ELISA for C3a in media exposed to treated cells (*n* = 10). (**C**) ELISA for C3a in lysates from cells (*n* = 7). Data shown are mean ± SEM. * *p* < 0.05, ** *p* < 0.01 determined by unpaired *t*-test.

**Figure 5 cells-11-02075-f005:**
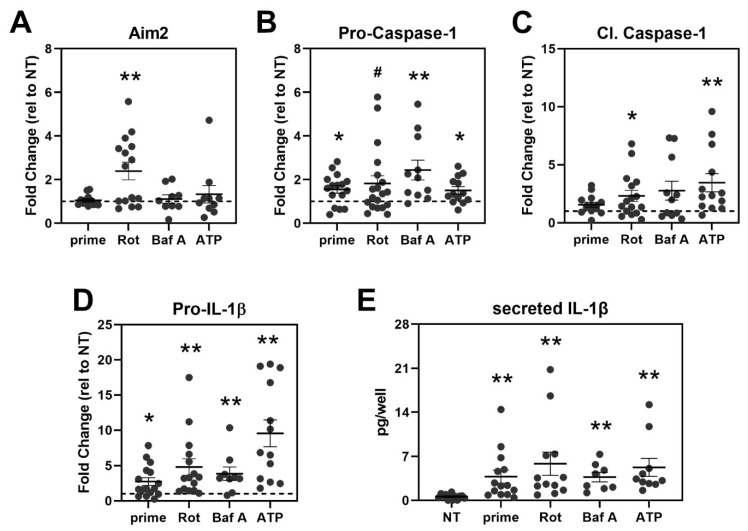
Activating treatments increased the content of Inflammasome components and downstream products in haRPE cells. Protein content analysis of Aim2 (**A**), Pro-Caspase-1 (**B**), Cleaved Caspase-1 (**C**), and Pro-IL-1β (**D**) after treatment. Protein levels were normalized to β-Actin (loading control) and plotted as fold change relative to no treatment (NT = 1, dashed line) for each donor. (**E**) Concentrations of secreted IL-1β in exposed cell culture media were determined by ELISA using a standard curve. The bar and whiskers represent the mean ± SEM. Each data point represents results from an individual donor. NT = no treatment, prime = LPS and IL-1α only, Rot = Rotenone, Baf A = Bafilomycin A. # *p* < 0.1, * *p* < 0.05, ** *p* < 0.01 were statistically different from untreated controls, as determined by one-sample *t*-tests in (**A**–**D**) and unpaired *t*-tests in (**E**).

**Figure 6 cells-11-02075-f006:**
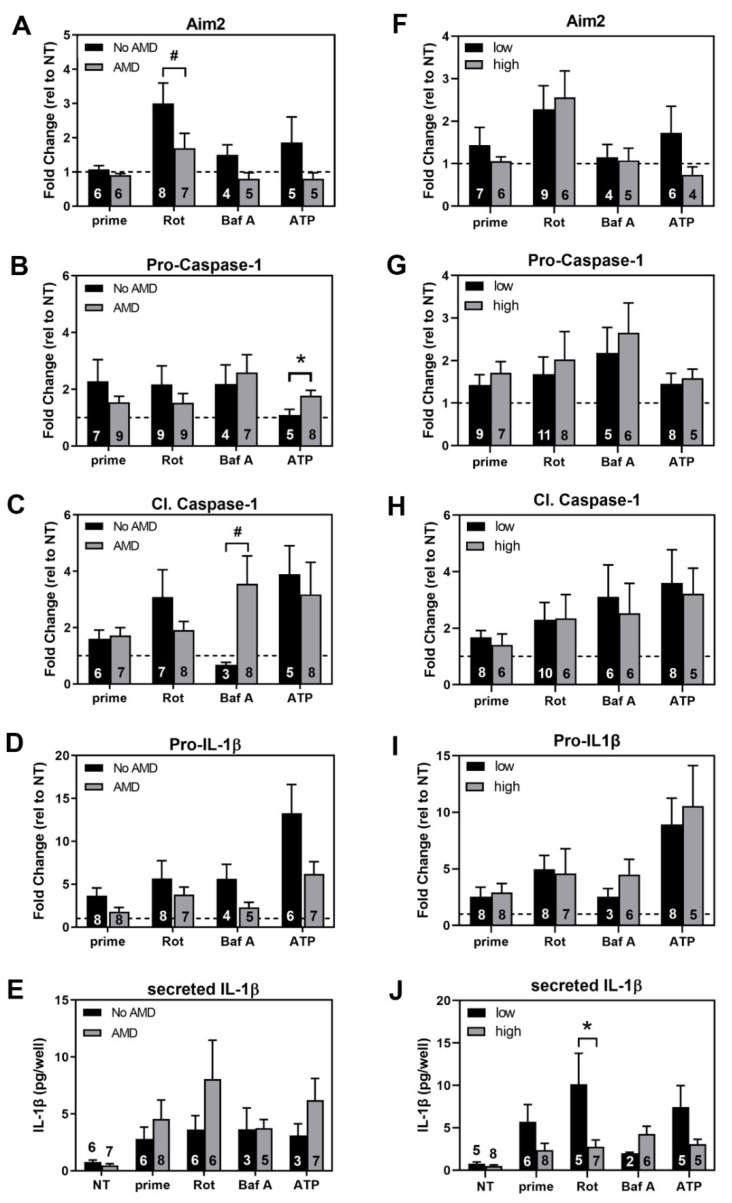
Comparison of Inflammasome Activation in haRPE cells by disease state and genotype. The protein content of Aim2, Pro-Caspase-1, Cleaved Caspase-1, and IL-1β were compared in cells from donors with AMD to donors without AMD (**A**–**E**), and compared in cells from donors with CFH high-risk genotype to a low-risk high-risk (**F**–**J**). Protein levels were normalized to β-Actin (loading control) and plotted as fold change relative to their treatment controls. Concentrations of secreted IL-1β from exposed cell culture media were determined by ELISA. Data shown are mean ± SEM. # denotes *p* < 0.1, * *p* < 0.05, determined by unpaired *t*-test. Numbers within bars are the number of individual donors used in each assay.

**Table 1 cells-11-02075-t001:** Mass spectrometry identification of complement and inflammasome proteins in RPE ^A^.

Protein Name	Uniprot ID	# Unique Peptides ^B^ (Average)	% Sequence Coverage ^C^ (Average)
Complement component 1 Q subcomponent-binding protein (C1QBP)	Q07021	11/12	62/62
Complement C1q subcomponent subunit B	P02746	0/4	0/22
Complement C1q subcomponent subunit C (C1QC)	P02747	0/3	0/15
Complement C1q tumor necrosis factor-related protein 5 (C1QTNF5)	Q9BXJ0	5/6	27/34
Complement component 3 (C3)	P01024	59/51	48/44
Complement component 4-A (C4A)	P0C0L4	54/56	40/43
Complement component 4-B (C4B)	P0C0L5	55/57	42/45
Complement component 5 (C5)	P01031	N.D.	N.D.
Complement component 6 (C6)	P13671	26/28	34/41
Complement component 7 (C7)	P10643	19/25	35/46
Complement component 8 alpha chain (C8A)	P07357	11/9	27/27
Complement component 8 Beta chain (C8B)	P07358	9/12	21/28
Complement component 8 gamma chain (C8G)	P07360	8/8	49/49
Complement component 9 (C9)	P02748	22/23	45/47
Complement Factor B (CFB)	P00751	2/6	4/12
Complement Factor H (CFH)	P08603	29/32	33/36
Complement Factor H-related protein 1 (CFHR1)	Q03591	9/10	41/38
Complement Factor H-related protein 2 (CFHR2)	P36980	5/5	31/27
Complement Factor H-related protein 3 (CFHR3)	Q02985	2/0	6/0
Complement Factor H-related protein 5 (CFHR5)	Q9BXR6	14/10	34/28
Complement Factor I (CFI)	P05156	14/13	29/28
Complement decay-accelerating factor (CD55)	P08247	8/7	29/29
MAC inhibitory protein (CD59)	P13987	4/4	25/25
Membrane cofactor protein (CD46)	P15529	9/8	21/21
Interleukin-6 (IL6)	P05231	N.D.	N.D.
Interleukin-6 receptor subunit beta (IL6ST)	P40189	11/11	21/21
Monocyte chemoattractant protein-1 (MCP1)	Q6UZ82	N.D.	N.D.
Interferon-inducible protein (AIM2)	O14862	N.D.	N.D.
NLR family member X1 (NLRX1)	Q86UT6	17/21	26/33
NACHT, LRR and PYD domains-containing protein 13 (NLRP13)	Q86W25	2/0	3/0
NACHT, LRR and PYD domains-containing protein 2 (NLRP2)	Q9NX02	N.D.	N.D.
NACHT, LRR and PYD domains-containing protein 3 (NLRP3)	Q96P20	N.D.	N.D.
NLR family CARD domain-containing protein 4 (NLRC4)	Q9NPP4	N.D.	N.D.
Apoptosis-associated speck-like protein containing a CARD (PYCARD)	Q9ULZ3	N.D.	N.D.
Caspase-1	P29466	N.D.	N.D.
Interleukin-1 beta (IL1B)	P01584	N.D.	N.D.

^A^ Organelle-enriched fraction prepared from RPE tissue for No AMD (MGS1, *n* = 32) and AMD (MGS2, *n* = 45) donors. ^B^ The number of peptides containing different amino acid sequences, regardless of any modification, that are attributed to a single protein/protein group. (# for MGS1/# for MGS2). ^C^ Provides the percentage of amino acids in the whole protein sequence that was identified in the sample. (% for MGS1/% for MGS2). N.D. = not detected.

## Data Availability

Data used to support the findings in this study are contained within this article and Appendix A. Data generated from mass spectrometry are available via ProteomeXchange with the identifier PXD033413.

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
