# Peer review of "Inflammasome Activation in Retinal Pigment Epithelium from Human Donors with Age-Related Macular Degeneration"

_cells, 2022, doi:10.3390/cells11132075_

Round 1
Reviewer 1 Report
In the article titled “Inflammasome activation in retinal pigmented epithelium from human donors with age-related macular degeneration,” by Ebling et al., the authors investigated inflammasome pathways in human RPE cells from samples with CFH risk factor and with/without AMD.
Major Comments:
- Introduction is long. Figure 1 needs to be in the introduction—not results. It would also all the authors to cut down on text
- Figure S2 ought to be in the main paper—as it is essential to the rest of the data
- N=3 for figure 3 seems a bit low.
- Figure 4 is too complicated with the groups and the analyses. Please separate out the key groups
- Figure 5 is the key figure. Maybe eliminate Figure 4 since Figure 5 clarifies things
Reviewer 2 Report
Authors have described the inflammasome signaling in RPE cells from human donors. As authors mentioned, they could not find the significant difference at the basal condition. However the activation increased IL-1b as a downstream of inflammasome in RPE cells from human donors, but there were not difference among subgroups.
Results 3.1.; This is not "Results". It should be switched to other section.
Results 3.3; The high/low risk ratio in AMD and No-AMD is almost the same, which supposed to be different. Could you discuss about it?
Results 3.3; Could you provide the distribution of AMD grading in AMD samples?
L340; CD59 is not significant. Please clearly describe that it is a trend.
L399-401; The description can be misleading a little. There are not any significant increases in mtDNA (Figure S5). Authors cannot evaluate the rotenone treatment as a significant increase.
L506-518; As they mentioned, there may be a difference between early and late AMD. Could you compare early and late AMD in this study? In addition, a MGS4 sample used in figure S1 looks no difference in PCR compared to early stage. Could you explain the reason?
Round 2
Reviewer 2 Report
I do not have any further comments.